# Aseptic Abscess Syndrome in Rheumatoid Arthritis Patient

**DOI:** 10.3390/medicina58101354

**Published:** 2022-09-27

**Authors:** Agnieszka Owczarczyk-Saczonek, Marta Kasprowicz-Furmańczyk, Jakub Kuna, Paulina Klimek, Magdalena Krajewska-Włodarczyk

**Affiliations:** 1Department of Dermatology, Sexually Transmitted Diseases and Clinical Immunology, School of Medicine, Collegium Medicum, The University of Warmia and Mazury, Al. Wojska Polskiego 30, 10-229 Olsztyn, Poland; 2Department of Rheumatology, School of Medicine, Collegium Medicum, University of Warmia and Mazury, 10-900 Olsztyn, Poland

**Keywords:** aseptic abscess syndrome, rheumatoid arthritis, adalimumab

## Abstract

Aseptic abscess syndrome (AAS) is a rare, potentially life-threatening disorder, with numerous features of neutrophilic dermatoses. The main symptoms include aseptic abscess-like collections in internal organs (spleen, liver, lungs), lack of microbes (bacteria, viruses, or parasites) after an exhaustive search, ineffectiveness of antibiotics, and high sensitivity to corticosteroid therapy. AAS is characterized by the development of deep, inflammatory abscesses and systemic symptoms (weight loss, abdominal pain, fever, and leukocytosis). They may be associated with inflammatory bowel disease (IBD) and autoimmune diseases. The patient in this study is a 67-year-old man, suffering from rheumatoid arthritis (RA), with numerous purulent abscesses in the mediastinum, within the subcutaneous tissue above the extension surfaces of the joints, and on the dorsum of the hands. The lesions are accompanied by bone destruction. The patient was treated with prednisone 40 mg and adalimumab, which resulted in a quick reduction of inflammatory markers and clinical improvement, as well as the healing and absorption of abscesses. Despite COVID-19 infection, treatment with remdesivir, prednisone, and adalimumab was continued, with the complete resolution of the lesions. AAS is difficult to recognize, so practitioners have to be aware of this condition, especially in patients with RA.

## 1. Introduction

Aseptic abscess syndrome (AAS) is a rare, potentially life-threatening disorder, with numerous features of neutrophilic dermatoses. AAS is characterized by the development of deep abscesses and systemic symptoms (weight loss, abdominal pain, fever, and leukocytosis). The main characteristic elements include aseptic abscess-like collections in internal organs (spleen, liver, lungs); lack of microbes, bacteria, viruses, and parasites after an exhaustive search; and ineffectiveness of antibiotics, including antitubercular treatment. Rapid clinical improvement after corticosteroid therapy with or without additional immunosuppressive therapy, followed by rapid radiographic signs of abscess resolution, is an important feature. Most commonly, it can be associated with inflammatory bowel disease (IBD), especially Crohn’s disease, and other autoimmune diseases [1,2,3].

AAS shares numerous features with neutrophilic dermatoses and is currently classified into this group [1,2,4]. Genetic predisposing factors have been identified. Homozygous LPIN2 mutations were associated with the familial cases of chronic osteomyelitis (Majeed syndrome) and SAPHO syndrome. LPIN2 encodes lipin 2, which is involved in the apoptosis of neutrophils. In contrast, LPIN2 mutations are associated with increased inflammasome activity and the release of pro-inflammatory IL-1 [5]. The expected association with mutations in the NOD2/CARD15 gene has not been unequivocally confirmed. However, NOD2/CARD15 and other susceptibility genes may enhance the expression of AAS as a result of a combination of polymorphisms [3].

Histopathological examination is mainly aimed at excluding other conditions (e.g., deep mycoses), similar to other neutrophilic diseases. The subcutaneous tissue is affected by an abundant infiltration of mature PMNs in the center, with histiocytes and single giant cells present at the periphery. In the acute phase, neutrophils dominate, then the organization of the epithelial border with palisading histiocytes appears. Fewer neutrophils are observed throughout the late phase. They are surrounded by fibrous tissue with macrophages [2].

A case of an AAS patient during leflunomide treatment is presented in the paper. The examination of the patient was conducted according to the Declaration of Helsinki principles.

## 2. Case Presentation

A 67-year-old man with numerous abscesses in the mediastinum, within the subcutaneous tissue in the joints of the limbs and concomitant destruction of the wrist bone, was admitted to the Department of Dermatology, Sexually Transmitted Diseases and Clinical Immunology, for the diagnosis and treatment of the lesions. The patient had been initially admitted to the Rheumatology Department due to pain in the joints with a month history of erythematous and infiltrative lesions, which evolved into aseptic abscesses with high inflammatory indexes (CRP 371 mg/L, procalcitonin 57.36 ng/mL), but normal leukocytosis (WBC 4.46 G/L, neutrophilia 88%). The lesions were accompanied by fever of up to 38 °C and significant weakness, i.e., the patient did not move on his own. The patient’s past medical history revealed that his RA had been treated with methotrexate (25 mg/week) for 6 years. The treatment was discontinued due to its ineffectiveness and leflunomide 20 mg/d was included 3 months prior to hospital admission.

On admission, the following deviations in laboratory tests were found: CRP 121 mg/L, WBC 9.85 G/L, neutrophils 92.3%, and procalcitonin 1.4 ng/mL. In addition to high inflammatory markers, it was found that the patient had a positive count of anti-nuclear antibodies (1:1280) in the immunofluorescence method, with the presence of anti-Ds-DNA +++, anti-histone ++, and anti-NUC in the immunoblotting and positive cANCA+ 1:160 and pANCA+ 1:1280 in the immunofluorescence method. However, the patient did not meet the criteria for the diagnosis of systemic lupus erythematosus or (ANCA)-associated vasculitis. Antibiotic therapy (clindamycin, second- and third-generation cephalosporins, ciprofloxacin, and rifampicin) did not lead to clinical improvement.

The abscesses were located in the subcutaneous tissue near the joints of the wrists and the dorsal aspect of both hands, above the extension surfaces of the left elbow, both knee and ankle joints, and in the supraclavicular area with a diameter of up to 3–8 cm (Figure 1A,B). A CT scan of the thorax revealed encapsulated fluid areas and pleural effusions with a layer thickness of 17–22 mm. They were located subcutaneously and positioned tangentially to the sternal outline of the left clavicle end (Figure 2). The MRI of the abdominal cavity revealed no abscesses. As the disease progressed, the abscesses were incised and emptied. Up to 100 mL of purulent, dense, sterile discharge was evacuated.

We performed a biopsy of an abscess of the right dorsum of the hand. Histopathological examination showed sparse infiltrates of mononuclear cells in the skin around the vessels, and acute inflammatory infiltration in the subcutaneous tissue (mainly neutrophilic), with necrotic foci, penetrating into the subcutaneous connective tissue (Figure 3).

After the exclusion of bacterial background (sterile blood cultures, Quantiferon Gold, and genetic probe for TB negative) and the diagnosis of aseptic abscess syndrome, iv pulses of methylprednisolone (500 mg every day for 3 days), followed by prednisone 40 mg, were included in the treatment. Fever resolved, and the patient’s well-being improved. The formation of new abscesses was inhibited, especially in the mediastinum. Clinical improvement was accompanied by CRP reduction to 53 mg/L and procalcitonin to 1.4 ng/mL. According to literature recommendations, adalimumab 80 mg was included as the initiating dose, and 40 mg was continued every 2 weeks. CRP values further decreased (15.3 mg/L), neutrophilia normalized to 46%, and the abscesses healed and were absorbed (Figure 3). Due to significant anemia (Hgb, 7.6 g/L; RBC, 2.79 T/L; reticulocytes 9‰), the patient was administered 1 unit of red blood cell concentrate twice (Figure 4).

During the hospitalization, the patient was transferred to the Orthopedic Department for several days. He underwent a resection of the right distal ulnar epiphysis, removal of the necrotic tissue from the forearm, and wrist abscess fistula, with the coverage of the skin loss with Optilene mesh. The procedure was performed due to purulent inflammatory fistula of 1/3 of the distal forearm and right wrist with trophic skin loss, liquefactive soft tissue necrosis, and the dislocation of the ulnar head in the distal radioulnar joint (Figure 5A,B).

The patient was infected with SARS-CoV-2 (hospitalized in the COVID ward) and treated with remdesivir, without complications. Adalimumab 40 mg every 2 weeks was continued without the recurrence of AAS symptoms, with the low activity of rheumatoid arthritis symptoms (DAS28).

## 3. Discussion

The spectrum of neutrophilic dermatoses includes numerous clinical forms. Their common feature is related to the accumulation of neutrophils in the skin due to the increased activity of chemotactic factors, and not due to an infection or coexistence with systemic disorders [1]. André et al. proposed that neutrophilic dermatoses should be divided into three subgroups. The first group includes superficial, epidermal, and pustular diseases, such as Sneddon–Wilkinson disease and other pustules. The second one focuses on Sweet’s syndrome, characterized by skin infiltration, papules, and plaques. The third group includes deep dermatoses, causing abscesses, or ulcers, with the prototype of the ulcers being pyoderma gangrenosum [1]. Therefore, the differential diagnosis of individual forms is very difficult, and the coexistence of various clinical forms is frequently reported.

Sparse descriptions of the coexistence of AAS and RA are available in the literature. Be et al. presented a case of a 66-year-old woman with interstitial lung disease concomitant with RA. She developed aseptic abscesses on the limbs and in the lungs. Dermal and pulmonary lesions rapidly improved after the administration of prednisolone at a dose of 0.5 mg/kg/day [6]. Hafner et al. described a case of aseptic liver abscesses diagnosed in a 41-year-old patient with Cogan’s syndrome, i.e., a rheumatic disease mainly involving ocular and audiovestibular symptoms, which was effectively treated with prednisone (1 mg/kg body weight), infliximab (5 mg/kg body weight), and azathioprine (100 mg/day) [7]. Another case was described by Santa Lucia et al. A 56-year-old woman had RA concomitant with multiple myeloma. After a bone marrow transplant, she showed symptoms of pyoderma gangrenosum of the lower leg. She also had numerous abscesses in the lungs and in the pericardial, hepatic, splenic, and pancreatic regions. The lesions resolved after the use of prednisone, azathioprine, and canakinumab [8].

The correct diagnosis of AAS must be supported by an exclusion diagnosis. It is of great importance as it enables rapid inclusion of glucocorticoid therapy, which results in rapid improvement and does not expose the patient to long-term antibiotic therapy [1,2]. As regards our patient, despite high CRP and procalcitonin levels, based on clinical symptoms, we decided to include appropriate treatment, although the patient’s symptoms met the criteria for the diagnosis of Systemic Inflammatory Response Syndrome (SIRS), which is most commonly accompanied by severe infections (heart rate > 90/min, body temperature > 38 °C, leukocytosis > 12,000 cells/mm^3^) [9,10]. Systemic symptoms and laboratory indicators verified in the course of AAS may suggest SIRS or sepsis. Therefore, patients require multiple blood culture tests. Although several blood cultures were sterile in our patient, we were aware that only 30–60% of blood cultures are positive for sepsis [6]. In addition, obtaining blood cultures after the initiation of empirical treatment with antibiotics reduces the sensitivity of the test by about 50%, with pre-antibiotic cultures being positive [9]. Furthermore, infectious endocarditis was ruled out based on echocardiography and ECG. Therefore, the decision to include glucocorticosteroids is usually difficult, and improving the clinical condition of the patient is based on retrospection evidence.

AAS is most commonly associated with nonspecific inflammatory bowel disease (IBD), but it may be concomitant with other autoimmune diseases (recurrent polychondritis, spondyloarthropathy, Behçet’s disease, rheumatoid arthritis, lupus erythematosus, nodular arteritis), or autoinflammatory (acne, SAPHO) and neoplastic ones (myeloma, monoclonal gammopathy) [1,2]. However, about 60% of patients do not present with any concomitant diseases [2,11].

In our patient, rheumatoid arthritis was the concomitant disease, and it is unclear whether leflunomide could be a factor that additionally initiated the AAS process. Some authors described the paradoxical reaction of the occurrence of pustular psoriasis after the use of leflunomide. The reaction was also associated with the hyperactivity of neutrophils [12]. However, leflunomide inhibits their activity, which is manifested by a decrease in the number of neutrophils in the synovial fluid in patients with RA [13]. Drug-induced AAS has not been described until now.

Interestingly, the patient had a positive count of anti-nuclear antibodies (1:1280) in the immunofluorescence method, with the presence of anti-Ds-DNA+++, anti-histone++, anti-NUC in the immunoblotting, and positive cANCA+ 1:160 and pANCA+ 1:1280 in the immunofluorescence method. The patient did not meet the criteria for the diagnosis of systemic lupus erythematosus or (ANCA)-associated vasculitis. However, after a year of treatment, ANCA antibody control yielded a negative result, while the anti-nuclear antibody titer was still 1:1280 in the immunofluorescence method. No significant antibodies were detected in immunoblotting. In our viewpoint, the results indicate physiological transient autoimmunity as a defensive function of the immune system, the aim of which is to eliminate apoptotic residues during the breakdown of neutrophils [14,15]. However, as regards other described cases, apart from high inflammatory index rates, anti-neutrophil cytoplasmic antibodies (ANCA) were negative [6,8]. Considering the above-mentioned immunological test results, the patient is still monitored for the possible development of (ANCA)-associated vasculitis or connective tissue diseases. However, after more than 1.5 years of observation and treatment, additional disorders were not found, and RA showed minimal disease activity.

The treatment of AAS includes systemic corticosteroids used as first-line therapy, usually prednisone between 0.5 and 1 mg/kg/day. Sometimes, they are initially combined with intravenous methylprednisolone pulses. It results in the rapid resolution of fever and pain, along with a decrease in the indexes of inflammation in the blood [1,4]. Due to the low number of cases described, there are no precise recommendations for the gradual reduction of prednisone doses. It is advisable to include other immunosuppressants (azathioprine, cyclophosphamide, methotrexate, cyclosporine) or anti-TNF-alpha agents (adalimumab, infliximab) to enhance treatment or reduce corticosteroid doses [1,4]. Some research described the cases of the efficacy of anakinra, an IL-1 receptor antagonist; canakinumab, an IL-1β antagonist; and colchicine [1,6,8]. Treatment involving splenectomy did not meet expectations because a relapse usually occurred after surgery [1].

Notably, patients with AAS require long-term use of immunosuppressants and anti-TNF-α drugs, which increases the risk of infection with tuberculosis and non-tuberculous mycobacteria (Santa Lucia). It requires proper monitoring of the patient.

## 4. Conclusions

AA is difficult to recognize, so practitioners have to be aware of this condition. It would be advisable to establish the official diagnostic criteria proposed by André et al., which would facilitate faster diagnosis and the inclusion of suitable treatment [2]. The concept of this syndrome is provisional and will probably evolve in the future.

## Figures and Tables

**Figure 1 medicina-58-01354-f001:**
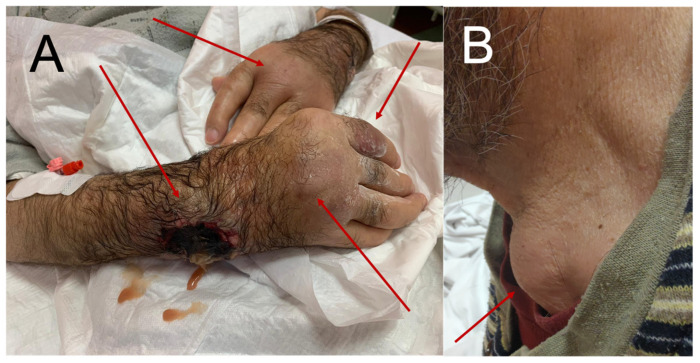
Abscesses in the subcutaneous tissue: (**A**) in the area of the hands, necrosis present in the area of the right wrist after the spontaneous rupture of an abscess; (**B**) abscess in the left supraclavicular fossa.

**Figure 2 medicina-58-01354-f002:**
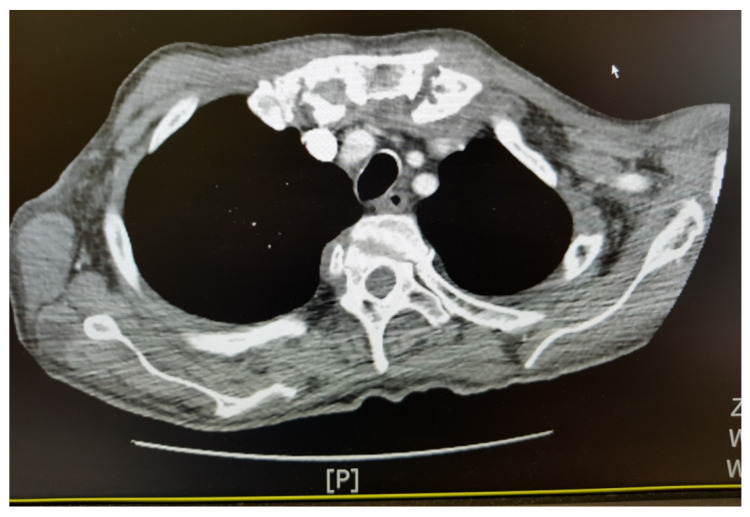
CT of the lungs: encapsulated fluid-containing lesions (lesion size: 26 × 17 mm and 16 × 12 mm) in the area of the subcutaneous tissue, positioned tangentially to the sternal outline of the left clavicle end—the overall image of the lesions is suggestive of abscesses; a significant degree of bilateral degenerative lesions in the area of the sternoclavicular joints; lymph nodes of the mediastinum, pulmonary hila and axillary fossa not enlarged; pleural fluid—the layer thickness up to 22 mm at the posterior wall on the left side and 17 mm on the right side; solitary, partially fibrous nodules visible in the right lung: 3 mm in diameter peripherally in 2P segment, positioned tangentially to the gap between segments 3P/6P and in seg. 6P, diameter 3 mm—post-inflammatory lesions?—for further inspection.

**Figure 3 medicina-58-01354-f003:**
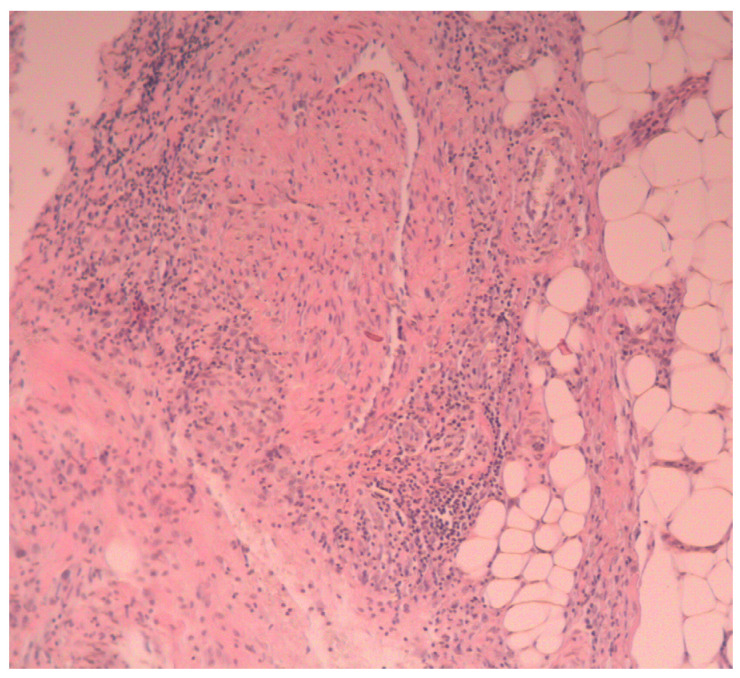
Histopathological examination: acute inflammatory infiltration in the subcutaneous tissue, mainly neutrophilic, with necrotic foci, penetrating into the subcutaneous connective tissue.

**Figure 4 medicina-58-01354-f004:**
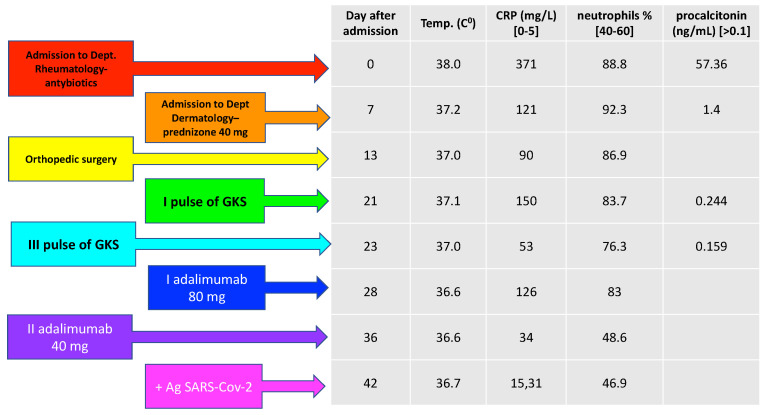
Laboratory findings and treatment in the course of hospitalization.

**Figure 5 medicina-58-01354-f005:**
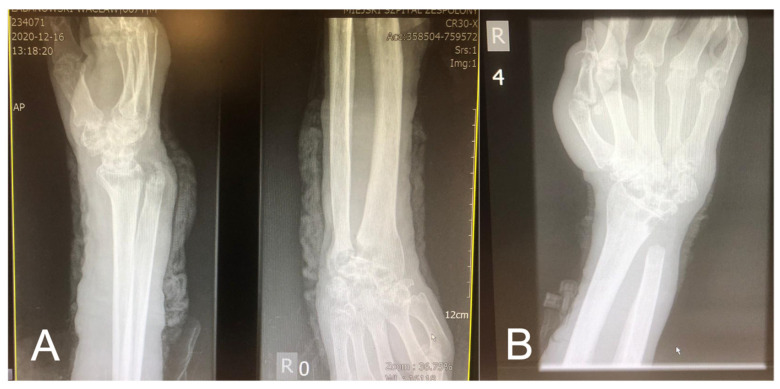
(**A**) X-ray of the right upper limb—subluxation of the wrist, contour deformity with a draining fistula of the head of the ulna; (**B**) X-ray after the resection of 1/3 of the distal right ulna.

## Data Availability

The analyzed data sets generated during the study are available from the corresponding author upon reasonable request.

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
