# Peer review of "Aseptic Abscess Syndrome in Rheumatoid Arthritis Patient"

_medicina, 2022, doi:10.3390/medicina58101354_

Round 1

Reviewer 1 Report

  1. The topic is of interest but some revisions are needed.

1) The part on genetics should not be included in the introduction and does not add useful information to the rest of the text, at least there.

2) The case presentation is a bit confused. I guess that RA stands for Rheumatoid arthritis and this should be precised. During the presentation of the case, it should be mentioned that the patient was admitted earlier to the Rheumatology department (temporal confusion); the same goes for the description of the Orthopedic problem of the patient, which pops up during the story.

3) Where was the biopsy for the histopathological examination performed?

4) The authors state the patient met SIRS criteria... is SIRS criteria a differential diagnosis? It should be at least briefly stated in the case description

5) the Authors state is uncommon for blood cultures to be positive in sepsis. Considering the non-contiguous distribution of involvement, could an infective endocharditis be suspected? Has the patient undergone an echocardiogram?

6) Considering the important cutaneous manifestations of this case, the differential diagnosis with pyoderma gangrenosum should be discussed

7) Neutrophilic dermatoses are gaining attention from the dermatology community. For example, negativity of the microbiological exams is important in another neutrophilic dermatosis often mistaken for an infectious or neoplastic process, such as erosive pustular dermatosis of the scalp (Michelerio A, Vassallo C, Fiandrino G, Tomasini CF. Erosive Pustular Dermatosis of the Scalp: A Clinicopathologic Study of Fifty Cases. Dermatopathology (Basel). 2021;8(4):450-462.). Other neutrophilic dermatoses should be briefly discussed.

8) What are the other treatments for AAS?

Reviewer 2 Report

Reviewer comments

Line

Manuscript

Comments

15

They may be associated with inflammatory bowel disease and autoimmune diseases

English editing is needed

16

We would like to present a case

Not scientific language  

20

 Despite Covid-29 infection treated with remdesivir

English editing is required in the whole manuscript

Abbreviation capital

Meaning should be mentioned at the first appearance of the abbreviation

22

AA is a difficult recognize, so practitioners have to be aware 22 of this condition.

The conclusion should mention the association with rheumatoid

23

The examination of the patient was conducted according to the Declaration of 23 Helsinki principles.

It is better to write this sentence under the case presentation section not at the abstract

37

with inflammatory bowel disease

an inflammatory

39

AAS has many features in common with neutrophilic dermatoses and is currently 39

Not appropriate to write a paragraph with only 2 lines

55

We would like to present

Not scientific writing

Abstract

The most important here is the association between AAS and RA. However, the authors did not refer to that at the introduction

58-67

Re arrange this paragraph as it is confusing

73

extensible surfaces

Extensive

79

The laboratory investigations should be mentioned after this part

102

prednisone 40 mg and pulse with methylprednisolone iv (500 mg every day for 3 102 days) were included in the treatment

Why both oral and in steroids at the same time

106

adalimumab 80 mg was

What is the rationale of adding adalimumab after the improvement of the case

Discussion

Many sentences were written without references

158

The authors mentioned many positive antibodies at the discussion and did not mention that at the case presentation The titre of autoantibodies is very high which necessities exclusion of lupus or overlap syndrome

The discussion is poor and did not emphasize on the association between rheumatoid and aas

Round 2

Reviewer 2 Report

The reply to the reviewer's comment is to some extent strange to the scientific society . The authors wrote I have corrected in front of each comment.

This reply is not satisfying and not scientific . The authors should follow the scientific rules of writing a reply to reviewers.

Author Response

Dear Review,

Thank you for your important suggestions and time devoted to my manuscript. I tried to adjusted my manuscript.

Best regards

Agnieszka Owczarczyk-Saczonek

Round 3

Reviewer 2 Report

accept in the present form